# QUALITY OVER QUANTITY: SEMI-SUPERVISED DETECTION OF ILLICIT BITCOIN FLOWS VIA FEATURE ENGINEERING

## ABSTRACT

Detecting illicit cryptocurrency transactions is hampered by extreme class imbalance, adversarial obfuscation, and a scarcity of reliable labels. While semi-supervised learning (SSL) offers a promising solution by leveraging unlabeled data, we show that its success is not guaranteed by data volume alone but is contingent on data quality. We introduce an SSL framework for identifying illicit flows in Bitcoin's Shared Send Mixers (SSMs) and make three contributions: (1) The first complete historical dataset of 163 million Bitcoin transactions with SSM classification; (2) Novel, high-fidelity features–KeyLinker address clustering and Shared Send Untangling (SSU) complexity metrics–designed to capture mixing structures and improve data quality; (3) A demonstration that SSL effectively leverages unlabeled data (F1-score: 0.84) precisely when guided by these quality-focused features. Crucially, we prove that common heuristics like One-Time Change (OTC), though abundant, introduce noise, while strategic reliance on higher-fidelity features like KeyLinker is essential. Our work establishes that in blockchain forensics, the path to better performance lies in smarter feature engineering for data quality, not just larger datasets.

**Keywords:** Blockchain, Bitcoin, Shared Send Mixer, Semi-Supervised Learning

## 1 INTRODUCTION

Bitcoin's decentralized architecture provides users with pseudonymity through cryptographic addresses, enabling financial autonomy without intermediaries. While this design upholds privacy principles, it has inadvertently facilitated illicit activities including money laundering, terrorist financing, darknet markets, and *scam* operations. According to Chainalysis (2025) Crypto Crime Report, illicit cryptocurrency addresses generated $40 billion in 2024, representing 0.14% of total network transactions. This persistent misuse underscores the critical need for effective blockchain forensic methods.

The Unspent Transaction Output (UTXO) model forms Bitcoin's transactional backbone Nakamoto (2008); Delgado-Segura et al. (2019); Lipton & Treccani (2021), where each transaction consumes existing outputs and creates new ones. Like physical banknotes, users must provide inputs covering both payment amount and miner fees, enabling privacy techniques while complicating tracing efforts. This model enables privacy-enhancing techniques like CoinJoin Maxwell (2013) while simultaneously complicating transaction tracing.

CoinJoin, a prominent transaction-mixing protocol introduced in 2013, exemplifies the dual-use challenge of privacy technologies. By aggregating multiple payments into a single transaction, it severs observable links between senders and receivers through input-output obfuscation. While serving legitimate privacy needs, this Shared Send Mixer (SSM) technique is weaponized by criminals to conceal illicit fund flows from wash trading, darknet markets, and ransomware operations European Union Agency for Law Enforcement Cooperation (2020; 2021). The computational hardness of untangling these transactions Atlas (2014); Yanovich et al. (2016) creates analytical blind spots for law enforcement.

Existing detection methodologies show promise yet face fundamental limitations. While graph neural networks (GNNs) and ensemble methods achieve over 90% accuracy in conventional flows, these supervised approaches require extensive labeled datasets–a critical barrier for analyzing mixed transactions due to CoinJoin's inherent complexity and the scarcity of reliable ground truth. This creates a fundamental impasse for supervised learning: the most complex and consequential transactions (mixed flows) have the least available reliable ground truth, making them a quintessential challenge for semi-supervised and weakly-supervised methods. This creates a paradox: the transactions requiring the most scrutiny have the least reliable labels, suggesting that the prevailing focus on acquiring more data must be complemented by a focus on improving the quality of the data we have. We acknowledge that off-chain labeling sources may introduce inaccuracies in illicit transaction classification (particularly for nuanced activities like scam operations), but prioritize transparent replication through publicly verifiable data. Semi-supervised learning presents a compelling alternative by leveraging both limited labeled data and abundant unlabeled records, as demonstrated in financial fraud detection Yin & Vatrapu (2017) and network anomaly analysis Zhang et al. (2020).

This study advances CoinJoin transaction forensics through three primary contributions, reframing the problem from one of data quantity to data quality:

1. **The Foundation: A Comprehensive Dataset.** We provide the raw quantity: the first complete historical dataset of CoinJoin transactions through synergistic integration of on-chain analysis and off-chain metadata spanning Bitcoin's entire history.

2. **The Enabler: Novel Forensic Features.** We introduce the tools to extract quality from quantity: KeyLinker Smolenkova & Yanovich (2025), an address clustering technique leveraging cryptographic key reuse patterns, and enhanced Shared Send untangling metrics Larionov & Yanovich (2023) specifically designed to decode mixed transaction structures and generate high-fidelity signals.

3. **The Proof: A Quality-Driven Semi-Supervised Framework.** We demonstrate that a semi-supervised learning framework outperforms supervised baselines by leveraging unlabeled data strategically. Crucially, we show that its success is contingent on the quality of features (e.g., KeyLinker vs. OTC) rather than the sheer volume of pseudo-labels, proving that performance is driven by data quality.

The remainder of this paper is structured as follows: Section 2 examines Bitcoin's UTXO transaction model and key anonymization techniques. Section 3 analyzes existing blockchain forensic approaches and CoinJoin detection challenges. Section 4 formally defines the illicit transaction identification problem and evaluation framework. Section 5 details our three-phase approach combining transaction clustering, feature engineering, and semi-supervised learning. Section 6 presents comparative results across multiple detection paradigms. We conclude with policy implications and future research directions in Section 7.

## 2 BACKGROUND: BITCOIN ANONYMIZATION TECHNIQUES

### 2.1 TRANSACTION MODEL

Bitcoin operates under a UTXO (Unspent Transaction Output) model, where each transaction consumes previous outputs as inputs and produces new outputs. Each output is associated with a script defining the conditions for spending. This design facilitates transaction chaining and allows for flexible ownership and payment schemes. However, the visibility of all transactions on the public blockchain also means that the flow of funds can be observed and analyzed.

As shown in Figure 1, the UTXO model's inherent transparency enables three principal privacy leakage vectors: address reuse across transactions, wallet fingerprinting through deterministic address generation patterns, and metadata exposure via spending timing analysis. These vulnerabilities have spawned various obfuscation techniques, creating an ongoing arms race between privacy-seeking users and blockchain analysts.

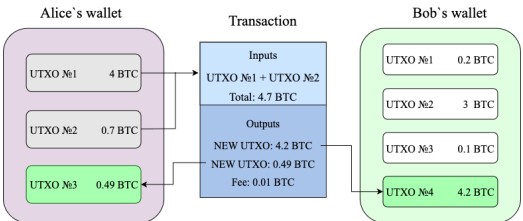

Figure 1: Bitcoin UTXO transaction model. Each transaction consumes previous outputs and creates new ones, enabling traceability but also exposing privacy leaks (e.g., address reuse, timing analysis).

## 2.2 ADDRESS CLUSTERING HEURISTICS

Despite the pseudonymous nature of Bitcoin addresses, certain heuristics make it possible to infer when multiple addresses are likely controlled by the same entity (Figure 2). The most widely used is the **Common Spending** (CS) heuristic, which assumes that if several addresses appear together as inputs in a transaction with a single output, they must belong to one user–since signing requires access to the corresponding private keys.

A second, equally influential method is the **One-Time Change** (OTC) heuristic. In a typical transaction, one output represents the actual payment while another returns change to the sender. If this change address is used only once, it provides a strong clue about wallet ownership and behavior.

These heuristics underpin most clustering techniques and have been validated in academic literature and blockchain analytics platforms.

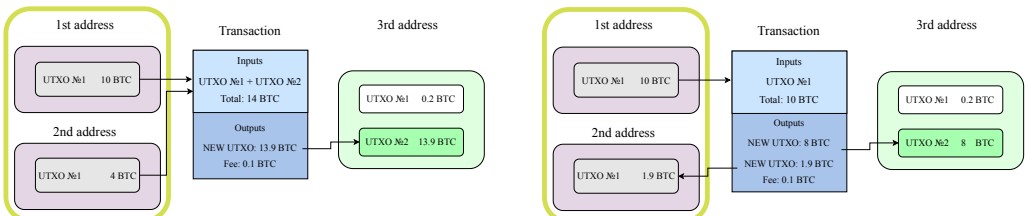

(a) Example of address clustering using CS heuristic. 1st address and 2nd address are owned by a single wallet.

(b) Example of address clustering using OTC heuristic. 1st address and 2nd address are owned by a single wallet.

Figure 2: Comparison of address clustering heuristics.

## 2.3 SHARED SEND MIXER TRANSACTIONS

Shared Send refers to a class of anonymization techniques based on the CoinJoin concept. In CoinJoin, multiple users collaboratively create a single transaction where inputs and outputs are pooled together. This makes it difficult to determine which output belongs to which input, thus obfuscating the flow of funds.

A Shared Send transaction typically features many inputs and multiple outputs of the same denomination. These transactions are often constructed using special-purpose wallets or services (e.g., Wasabi Wallet) designed to facilitate anonymity.

Such transactions appear organically on the blockchain due to growing user adoption of privacy tools. However, they can also be used by illicit actors to obfuscate traces of illegal activity, such as darknet market payments or ransomware.

Despite their goal of anonymity, Shared Send transactions are subject to partial deanonymization. For example, when not all output values are identical, it becomes easier to determine the relationship between input and output data. Alternatively, users participating in multiple CoinJoins with similar behavior may be grouped together.

Within the SSU framework Larionov & Yanovich (2023), transactions fall into five categories. Some are **regular**, with too few inputs or outputs (less than two) to require untangling. Others are **simple**, where the mapping from inputs to outputs is uniquely identifiable. More complex cases may be **separable**, neatly divided into separate, non-overlapping subgroups of senders and receivers. Still others remain **ambiguous**, where multiple plausible mappings exist between inputs and outputs, or even **time-limited**, where the computational effort required to untangle them is prohibitively high.

Understanding these patterns is crucial for robust detection of anonymization schemes and building resilient forensic models.

## 3 RELATED WORK

Since its inception, privacy preservation has been one of the main advantages of the Bitcoin blockchain Nakamoto (2008). This system allows its users the ability to carry out transactions directly between participants without intermediaries, enhancing privacy. The participants of the network are hidden behind pseudonymous addresses, which are not directly related to real identities. However, these features also create a favorable environment for illicit activities, including money laundering, terrorist financing, and illicit trade.

While user addresses are pseudonymous, the public availability of data on all transactions provides an opportunity to analyze it, enabling its utilization for research. Early research in this sphere examined the privacy of Bitcoin network users Androulaki et al. (2013) and the potential for conducting deanonymization through topological analysis of the transaction graph Vallarano et al. (2020), thus illustrating the complex balance between anonymity and transparency.

CoinJoin Maxwell (2013) significantly enhances anonymity by combining transactions from multiple users into a single transaction, making it difficult to trace transaction inputs and outputs. However, this same feature can be exploited by criminals to obfuscate the origins and distribution of illicit funds. Studies on CoinJoin and 'Shared Send' transactions Yanovich et al. (2016); Larionov & Yanovich (2023; 2024) demonstrate the inherent complexities in deconstructing mixed transactions, complicating differentiation between privacy-seeking users and criminals.

In parallel, address clustering–the process of linking pseudonymous blockchain addresses to real-world entities–has evolved considerably, transitioning from early heuristic-based techniques to sophisticated machine learning-driven methodologies Ermilov et al. (2017); Möser & Narayanan (2022); Liu et al. (2023), significantly improving the accuracy of detecting concealed links. Recent advancements include semi-supervised graph neural networks (GNNs), trained on a dataset of 13 million transactions, have achieved a remarkable 92% accuracy in binary classification of illicit activity Nerurkar (2022). Similarly, gradient-boosted ensemble models have demonstrated exceptional performance, successfully categorizing users into 16 distinct classes (e.g., darknet markets, mixing services) with an accuracy of 91% Nerurkar et al. (2021).

Mixing services specifically aim to obscure fund flows. Initial detection relied on statistical and heuristic methods. With advancements in machine learning and graph analysis, identifying these mixers became more efficient. A notable example includes decision trees that have been optimized via reduced-error pruning, which can detect an impressive 97% of mixing services while relying on just 8 key transaction features, such as activity frequency and UTXO age Rathore et al. (2022). Comparative studies of various classification algorithms (Decision Trees, Random Forest, SVM) show that ensemble methods like Random Forest often achieve high accuracy (up to 90%) in detecting suspicious transactions Alarab et al. (2020). Systematic reviews report overall recognition accuracies of up to 87% Lin et al. (2022).

Deep neural networks also demonstrate high accuracy in detecting hidden patterns distinguishing regular transactions from those involving mixing services Yin & Vatrapu (2017); Nan & Tao (2018). Recent innovations include metapath-aware graph neural networks that encode heterogeneous transaction features, demonstrated a 7% improvement in money laundering detection precision compared to GNNs Song & Gu (2023), and hypergraph-based models like CENSor, hypergraph-based model that integrates Cluster-GCN embeddings with Random Forest classifiers to achieve robust illicit transaction detection Lee et al. (2024). Furthermore, advanced clustering techniques have proven particularly valuable for uncovering organized criminal networks involved in money laundering op-

erations. These methods can identify criminal communities within blockchain transaction graphs and reveal key nodes that frequently interact with CoinJoin transactions Wahrstätter et al. (2023).

A particularly promising line of research in blockchain analytics has focused on enhancing the efficiency and accuracy of Bitcoin address classification through novel feature selection methodologies. Among recent innovations in this field, the paper Sie et al. (2024) proposes a feature selection method that combines quantum computation principles with classical machine learning. By leveraging quantum-inspired algorithms, the authors achieve state-of-the-art results for classifying illicit and licit addresses on large Bitcoin datasets, further underlining the value of feature engineering and dimensionality reduction in transaction forensics.

## 4 PROBLEM STATEMENT

We formulate the illicit transaction detection task as a binary classification problem over Bitcoin transactions. Let $\mathcal{T}$ denote the universe of all Bitcoin transactions. Our goal is to learn a classifier $f : \mathcal{T} \to \{0, 1\}$, where $f(t) = 1$ indicates that transaction $t \in \mathcal{T}$ is illicit (e.g., associated with mixing services, darknet markets, or scams), and $f(t) = 0$ otherwise.

Each transaction $t \in \mathcal{T}$ is represented through its native UTXO structure:

- $\mathcal{I}_t = \{(a_n, A_n)\}_{n=1}^N$: Input UTXO multiset, where $a_n \in \mathbb{R}_{\geq 0}$ is the scalar input amount and $A_n \in \mathcal{A}$ is the source address

- $\mathcal{O}_t = \{(b_m, B_m)\}_{m=1}^M$: Output UTXO multiset, where $b_m \in \mathbb{R}_{\geq 0}$ is the output amount and $B_m \in \mathcal{A}$ is the destination address.

Addresses carry semantic tags from external sources and clustering heuristics:

$$\text{Tag} : \mathcal{A} \to (\mathcal{L} \cup \{\bot\}) \times (\mathcal{C} \cup \{\bot\}),$$

where $\mathcal{L} = \{\text{exchange}, \text{mixer}, \text{darknet}, \text{gambling}, \ldots\}$ are entity labels, $\mathcal{C} = \{\text{illicit}, \text{licit}\}$ are legitimacy labels, and $\bot$ indicates missing labels. Tags propagate through clustering relationships ($\sim$):

$$\forall A, A' \in \mathcal{A} : A \sim A' \implies \text{Tag}(A) = \text{Tag}(A')$$

The clustering relationship is established by KeyLinker public key associations, CS and OTC heuristics.

Bitcoin transactions may contain repeated addresses in their inputs and outputs–a potentially useful characteristic for classification. We preserve this raw UTXO structure while enabling complexity analysis through strategic simplification Yanovich et al. (2016): $t \mapsto t_{\text{sim}} = \texttt{Simplify}(t)$. This mapping groups UTXOs by addresses and their clustering relationships exclusively to determine the transaction's untangling class $\kappa(t) \in \{\text{regular}, \text{simple}, \text{separable}, \text{ambiguous}, \text{time-limit}\}$ and untangling-related features. The $\kappa(t)$ classification feeds into the feature engineering pipeline as critical SSU attributes, while the original address repetitions remain preserved in $\mathcal{I}_t$ and $\mathcal{O}_t$ for feature extraction.

## 5 METHODOLOGY

Our methodology is designed to identify illicit CoinJoin transactions in the Bitcoin blockchain by leveraging both supervised and semi-supervised learning techniques, enhanced by heuristic clustering and extensive feature engineering. Our methodological approach encompasses several key stages: dataset collection, labeling, feature engineering and semi-supervised classification modeling.

### 5.1 DATA COLLECTION, LABELING AND FEATURE ENGINEERING

The dataset includes Bitcoin blockchain data collected from the Bitcoin Core up to block 882,421 (dated February 6, 2025). To enhance our analysis, we integrated address labels from services including WalletExplorer, Elliptic++ Dataset, MBAL Dataset, and Kaggle datasets ChainToolAI; Garin, categorizing addresses by service type (exchanges, mixers, gambling, services, and mining pools) and their legality.

Table 1: Comprehensive dataset statistics.

| Transactions | | Addresses | |
|---|---|---|---|
| Total | 1,150.9M | Total | 1,370.1M |
| Labeled | 161.2M | Labeled | 39.0M |
| CoinJoin | 163.4M | KeyLinker | 131.4K |
| Labeled CJ | 4.6M | CS Heuristic | 859.0M |
| OTC Tx | 188.9M | OTC Heuristic | 472.3M |
| **SSU complexity classification (transactions)** | | | |
| Simple (SSU_1) | 99.1M | Separable (SSU_2) | 24.2M |
| Ambiguous (SSU_3) | 10.5M | Time-limit (SSU_4) | 5.4M |
| Regular (SSU_5) | 24.3M | | |
| **Legality labels** | | **Service categories** | |
| Illicit | 33.2K | Service | 18.2M |
| Legal | 251.1K | Exchange | 114.7M |
| | | Gambling | 13.2M |
| | | Mixer | 11.5M |
| | | Mining | 1.1M |

Our dataset comprises approximately 1.15 billion transactions, out of which 163 million are Coin-Join transactions, with 4.6 million explicitly labeled (Table 1). The dataset contains 1.37 billion unique Bitcoin addresses, includes 33,229 illegal and 251,083 legal addresses.

We manually resolved duplicates and conflicting labels, addressing ambiguities such as addresses tagged simultaneously as mixers and exchanges. Addresses were grouped using basic heuristic methods such as CS and OTC. However, a clustering approach based on the reuse of public keys, KeyLinker Smolenkova & Yanovich (2025), was also used. Upon acceptance, we will release our dataset.

For models training, we designed four groups of features. The first captures UTXO attributes, such as the average lifetime of outputs and the number of inputs and outputs. The second group focuses on transaction values, from basic sums and fees to more nuanced indicators like the market concentration index. The third group measures address-level behavior, for instance, whether addresses repeat across inputs and outputs. Finally, we extend the feature set with specialized attributes: SSU complexity labels and off-chain service associations (exchanges, miners, mixers, gambling and service).

Continuous features were normalized via StandardScaler, categorical features were represented by one-hot coding, and class imbalances were compensated using class weighting in the models.

## 5.2 THE DATA QUALITY PRINCIPLE FOR PSEUDO-LABELING

Contrary to the standard SSL approach of labeling all high-confidence predictions, we adopt a strategic approach informed by our feature analysis. We hypothesize that not all pseudo-labels are equally valuable; the quality of a pseudo-label is intrinsically linked to the quality of the features used to generate it. Specifically, we prioritize pseudo-labels derived from two sources of high-fidelity signal:

1. **Transaction Structural Quality**: Transactions that are more easily untangled (e.g., SSU Simple and Separable classes) provide cleaner structural patterns for the model to learn from, compared to Ambiguous or Time-Limited transactions.

2. **Clustering Heuristic Quality**: Pseudo-labels associated with addresses clustered by high-fidelity methods like KeyLinker (based on cryptographic proof) are more reliable than those from noisier heuristics like OTC.

This principle ensures our expanded training set is not just larger, but *better*, with a higher proportion of high-quality, reliable examples that enhance learning rather than introducing noise.

## 5.3 CLASSIFICATION FRAMEWORK

We partitioned the labeled dataset of 4.62 million CoinJoin transactions into training (80%), validation (10%), and test sets (10%), maintaining class proportions.

Given the high class imbalance illicit CoinJoin transactions constitute only about 12% of the labeled dataset–accuracy is not an appropriate performance measure. A trivial classifier that always predicts "legal" achieves high accuracy but no utility for forensic analysis.

Models trained included Random Forest, XGBoost, and CatBoost. Model performance was assessed using ROC AUC, Precision-Recall AUC, F1-score, precision, and recall metrics. We optimized for the F1-score to balance precision and recall.

We used stratified 5-fold cross-validation on the training set, with class weights set to balanced in all classifiers. Oversampling methods such as SMOTE or ADASYN were deliberately not applied, as pseudo-labeling later introduces new positive examples.

### Pseudo-labeling

We exploit the pool of unlabeled CoinJoin transactions through a selective pseudo-labeling scheme. The trained classifier is applied to the unlabeled transaction pool, and in each batch only the most confident predictions are retained. Rather than relying on fixed thresholds, we select the top fraction of samples on both sides of the decision boundary, adjusting the share of positives and negatives.

After collection, the pseudo-labeled dataset is merged with the original training data. The final expanded dataset is then used to retrain the model, extending its control without introducing excessive noise.

## 6 Numerical experiments

### 6.1 Experimental Platform

All experiments were conducted on a high-performance server configured with 200 GB RAM and Intel® Core™ i9-14900KF × 32 CPUs.

### 6.2 Supervised Training Phase

We first assess the effectiveness of our feature engineering and modeling approach in a fully supervised setting. The goal at this stage is to establish how well the available labeled data can distinguish illicit from licit CoinJoin transactions, and to benchmark a set of classifiers before incorporating unlabeled examples via pseudo-labeling.

Three model types were evaluated: XGBoost, CatBoost and Random Forest. To ensure fair comparison and optimal performance, we conducted stratified cross-validation for hyperparameter selection. This systematic model selection is the basis for all following experiments.

We evaluated each model on validation and hold-out datasets. Metrics included ROC AUC, precision, recall, F1-score values for classification (Table 2).

All models demonstrate a strong balance between true positive and true negative detection, with relatively low false positive rates.

The inclusion of REUSE (key reuse), CS (common spending) features leads to measurable gains in all performance metrics, confirming their critical importance for transaction forensics. Adding OTC features reduced metrics, while combining all features without OTC yielded the best results.

XGBoost achieves the best supervised performance with an F1-score of 0.845 (default+reuse+cs+ssu) and ROC-AUC = 0.970, closely followed by CatBoost (F1-score up to 0.830).

Recall is of great importance in this context: missing an illegal transaction is fraught with undetected criminal flows, while high accuracy is necessary to avoid overloading analysts due to false positives. The excellent F1-scores and balanced confusion matrices for the most efficient ensemble models demonstrate their ability to find this balance.

Table 2: Metrics by feature set for all models.

| Model | Features | | | | | Metrics | | | |
|---|---|---|---|---|---|---|---|---|---|
| | DEFAULT | REUSE | CS | OTC | SSU | Precision | Recall | F1-score | ROC AUC |
| | ✓ | | | | | 0.929 | 0.689 | 0.791 | 0.958 |
| | ✓ | ✓ | | | | 0.929 | 0.730 | 0.818 | 0.966 |
| | ✓ | ✓ | ✓ | | | 0.930 | 0.740 | 0.824 | 0.969 |
| CatBoost | ✓ | ✓ | ✓ | ✓ | | 0.928 | 0.740 | 0.823 | 0.967 |
| | ✓ | | | | ✓ | 0.926 | 0.705 | 0.800 | 0.960 |
| | ✓ | ✓ | ✓ | | ✓ | **0.936** | 0.746 | 0.830 | **0.970** |
| | ✓ | ✓ | ✓ | ✓ | ✓ | 0.930 | 0.745 | 0.827 | 0.968 |
| | ✓ | | | | | 0.875 | 0.762 | 0.814 | 0.959 |
| | ✓ | ✓ | | | | 0.888 | 0.790 | 0.837 | 0.967 |
| | ✓ | ✓ | ✓ | | | 0.897 | 0.796 | 0.844 | **0.970** |
| XGBoost | ✓ | ✓ | ✓ | ✓ | | 0.895 | 0.792 | 0.841 | 0.968 |
| | ✓ | | | | ✓ | 0.882 | 0.767 | 0.821 | 0.961 |
| | ✓ | ✓ | ✓ | | ✓ | 0.900 | **0.792** | **0.842** | **0.970** |
| | ✓ | ✓ | ✓ | ✓ | ✓ | 0.901 | 0.788 | 0.840 | 0.969 |
| | ✓ | | | | | 0.883 | 0.739 | 0.804 | 0.957 |
| | ✓ | ✓ | | | | 0.906 | 0.743 | 0.816 | 0.962 |
| | ✓ | ✓ | ✓ | | | 0.899 | 0.769 | 0.829 | 0.967 |
| RandomForest | ✓ | ✓ | ✓ | ✓ | | 0.908 | 0.739 | 0.825 | 0.960 |
| | ✓ | | | | ✓ | 0.893 | 0.731 | 0.805 | 0.957 |
| | ✓ | ✓ | ✓ | | ✓ | 0.901 | 0.769 | 0.830 | 0.967 |
| | ✓ | ✓ | ✓ | ✓ | ✓ | 0.907 | 0.744 | 0.818 | 0.962 |

## 6.3 SEMI-SUPERVISED LEARNING WITH PSEUDO-LABELING

While supervised models performed robustly, the vast pool of unlabeled CoinJoin transactions presents an opportunity for further improvement. Informed by our analysis that data quality is paramount (Section 5.2), we employ a selective pseudo-labeling scheme. The trained classifier is applied to the unlabeled transaction pool, and we retain only the most confident predictions, which are disproportionately found in the more tractable SSU complexity classes. This ensures the expanded training dataset has a higher proportion of 'quality' examples. Rather than relying on fixed thresholds, we select the top fraction of samples on both sides of the decision boundary, adjusting the share of positives and negatives.

As shown in Table 3, performance remained stable across models with F1-scores around 0.81–0.84 and ROC AUC values near 0.97. Crucially, the best results were consistently achieved with the `Default+REUSE+CS+SSU` feature set—the same combination identified as high-quality in supervised experiments. In contrast, adding the noisier OTC features degraded performance, even though it increased the number of pseudo-labels. This confirms that SSL gains depend not on dataset expansion alone, but on the quality of the features guiding pseudo-label selection.

XGBoost remained the most robust across both supervised and SSL settings, showing the smallest precision drop and stable F1-scores. CatBoost exhibited similar trends but with slightly lower precision, while Random Forest benefited least from pseudo-labeling, sometimes showing small degradations.

Pseudolabeling slightly increased recall (up to +0.03) while reducing precision (from -0.04 to -0.05). In practice, this means that the model detected more illegal transactions, but at the cost of introducing additional false positives. For forensic analysis, this compromise is often acceptable: recall is crucial to identify hidden illegal flows, while a small increase in the number of false positives can be handled by analysts.

The semi-supervised phase did not produce dramatic metric gains, but it reinforced our central claim that quality-focused features determine the effectiveness of SSL. When guided by reliable signals (KeyLinker, SSU), pseudo-labeling improves robustness; when expanded with noisy heuristics (OTC), additional data harms performance.

Table 3: Metrics by feature set for semi-supervised learning with pseudo-labeling.

| Model | Features | | | | | Metrics | | | |
|---|---|---|---|---|---|---|---|---|---|
| | DEFAULT | REUSE | CS | OTC | SSU | Precision | Recall | F1-score | ROC AUC |
| CatBoost | ✓ | | | | | 0.848 | 0.759 | 0.801 | 0.956 |
| | ✓ | ✓ | | | | 0.873 | 0.775 | 0.821 | 0.964 |
| | ✓ | ✓ | ✓ | | | 0.866 | 0.795 | 0.829 | 0.966 |
| | ✓ | ✓ | ✓ | ✓ | | 0.866 | 0.788 | 0.825 | 0.964 |
| | ✓ | | | | ✓ | 0.856 | 0.764 | 0.807 | 0.958 |
| | ✓ | ✓ | ✓ | | ✓ | 0.868 | 0.803 | 0.834 | 0.968 |
| | ✓ | ✓ | ✓ | ✓ | ✓ | 0.874 | 0.788 | 0.829 | 0.966 |
| XGBoost | ✓ | | | | | 0.865 | 0.757 | 0.807 | 0.957 |
| | ✓ | ✓ | | | | 0.891 | 0.779 | 0.832 | 0.966 |
| | ✓ | ✓ | ✓ | | | 0.887 | 0.796 | 0.839 | 0.969 |
| | ✓ | ✓ | ✓ | ✓ | | 0.892 | 0.787 | 0.836 | 0.966 |
| | ✓ | | | | ✓ | 0.873 | 0.763 | 0.814 | 0.959 |
| | ✓ | ✓ | ✓ | | ✓ | **0.897** | **0.797** | **0.845** | **0.969** |
| | ✓ | ✓ | ✓ | ✓ | ✓ | 0.890 | 0.787 | 0.836 | 0.967 |
| RandomForest | ✓ | | | | | 0.853 | 0.757 | 0.802 | 0.955 |
| | ✓ | ✓ | | | | 0.875 | 0.762 | 0.814 | 0.961 |
| | ✓ | ✓ | ✓ | | | 0.877 | 0.781 | 0.826 | 0.965 |
| | ✓ | ✓ | ✓ | ✓ | | 0.870 | 0.765 | 0.814 | 0.959 |
| | ✓ | | | | ✓ | 0.858 | 0.751 | 0.801 | 0.955 |
| | ✓ | ✓ | ✓ | | ✓ | 0.882 | 0.777 | 0.826 | 0.965 |
| | ✓ | ✓ | ✓ | ✓ | ✓ | 0.872 | 0.768 | 0.817 | 0.960 |

## 7 CONCLUSION

This work demonstrates that effective detection of illicit cryptocurrency transactions requires prioritizing data quality over data quantity. We have shown that simply acquiring more labeled data is insufficient–successful detection depends on strategic feature engineering to enhance data quality, particularly in complex domains like blockchain forensics where reliable labels are scarce.

Our novel features, including the KeyLinker clustering technique based on cryptographic key reuse patterns and the Shared Send Untangling complexity metrics, provided the means to measure and improve data quality. These high-fidelity features significantly outperformed traditional heuristics, confirming that feature quality substantially outweighs feature quantity in illicit transaction detection. Our semi-supervised learning framework further proved that models trained on strategically expanded high-quality data outperform those trained on larger, noisier datasets.

These findings advance blockchain forensic methodology by establishing that gradient-boosted models, particularly XGBoost, provide the most robust performance for capturing Bitcoin's complex transaction patterns. More importantly, we demonstrated that quality-aware semi-supervised learning successfully leverages Bitcoin's inherent pseudonymity to overcome label scarcity, but only when guided by high-fidelity features rather than simple confidence thresholds.

This work establishes a foundation for next-generation blockchain forensics that balances effective illicit flow detection with respect for legitimate privacy interests. Future work should develop more advanced quality assessment metrics, explore noise-resistant learning architectures, and implement real-time quality evaluation systems for blockchain-scale analysis. By shifting the focus from data quantity to data quality, our approach opens new pathways for effective analysis in challenging, adversarial domains.

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
