# OpenReview forum: "Quality over Quantity: Semi-Supervised Detection of Illicit Bitcoin Flows via Feature Engineering"
_ICLR.cc/2026/Conference — Submitted to ICLR 2026_

### Official Review · Reviewer_Y9sg · 2025-10-27

**Soundness:** 2
**Presentation:** 2
**Contribution:** 1
**Rating:** 2
**Confidence:** 5

**Summary:**

The paper studies the detection of illicit Bitcoin transactions and demonstrates the effectiveness of two new features, i.e., a clustering technique based on cryptographic key reuse pattens (KeyLinker, published in 2025) and the Shared Send Untangling complexity metrics (SSU, published in 2023). With experiments on a newly constructed dataset, these two features outperforms traditional heuristics for illicit transaction detection, especially the often used One-Time Change (OTC) features even decreasing detection performance.

**Strengths:**

The main strength of the paper is introducing the above two features for illicit transaction detection in Bitcoin.

**Weaknesses:**

The contributions of this paper are rather limited, and somehow I feel that this paper is a bit out of the scope of ICLR focusing on learning representation. The paper uses feature engineering to improve the performance of a few ML models (CatBoost, XGBoots, and RF). However, even those two effective features are not newly proposed in this paper.

The paper does not present the details of these two features, and the analysis in the end of Section 6 is rather simple and does not lead to any deep insights.

Section 3 can be better structured. After reading the related work, I cannot see how this paper will differ from them and what will be its new contributions.

**Questions:**

There are a few other types of approaches and features discussed in Section 3. The experimental evaluation can be improved by more comparison.

---

### Official Review · Reviewer_a5q8 · 2025-10-31

**Soundness:** 1
**Presentation:** 2
**Contribution:** 1
**Rating:** 0
**Confidence:** 5

**Summary:**

The paper presents a semi-supervised learning framework for identifying illicit flows in Bitcoin;s shared send mixers (SSMs). The main claims are the classification of 163 millions of Bitcoin transactions, the the features-keyLinker address clustering and the experimental results showing quality focused features can enhance semi-supervised learning performance.

**Strengths:**

The problem of detection of illicit Bitcoin transaction flows is important and challenging especially in the presence of large unlabeled data.

The use of feature engineering to obtain quality features for high performance semi-supervised learning is widely recognized as a good practice.

The related work covers some of recent Bitcoin literatures.

**Weaknesses:**

The paper needs to be significantly improved in both technical contributions and presentation. Here are some suggestions:

(1) The paper makes three claims: the classification of 163 millions of Bitcoin transactions, the features-keyLinker address clustering and the experimental results showing quality focused features can enhance semi-supervised learning performance on unlabeled Bitcoin flows. However, neither of the three claims are new in both concepts and technical development. The authors seem unaware of large literature and open source efforts in the detection of Illicit Bitcoin transactions. One example is the bulk of work based on the Elliptic dataset, e.g.,  the Elliptic++ (ACM SIGKDD 2023), which demonstrates the feasibility of detection of illicit Bitcoin transactions and illicit Bitcoin addresses with only small limited label data, with feature engineering, feature clustering and ensemble learning.

(2) The paper lacks technical novelty in either data collection stage or machine learning for detection of illicit Bitcoin flows.

(3) The experimental evaluation is poor and lacks the comparison to the state of the art methods for detection of illicit Bitcoin transactions/addresses.

**Questions:**

See the weakness section.

---

### Official Review · Reviewer_FdG4 · 2025-11-01

**Soundness:** 2
**Presentation:** 2
**Contribution:** 2
**Rating:** 2
**Confidence:** 3

**Summary:**

The paper presents a semi-supervised learning (SSL) framework to detect illicit transactions in Bitcoin's Shared Send Mixers (SSMs), with three contributions: a historical dataset of 163 million Bitcoin transactions labeled for SSMs; novel high-fidelity features such as KeyLinker address clustering and Shared Send Untangling (SSU) complexity metrics that capture mixing structures and improve data quality; and an empirical demonstration that SSL, when guided by these quality-focused features, effectively leverages unlabeled data achieving an F1 score of 0.84.

**Strengths:**

S1. The authors have released the first complete historical dataset of CoinJoin transactions, integrating on-chain analysis with off-chain metadata across Bitcoin’s entire history.

S2. The paper introduces novel forensic features - KeyLinker address clustering and enhanced Shared Send untangling metrics - that extract high-fidelity signals by decoding mixed transaction structures.

S3. The work shows that a semi-supervised framework outperforms supervised baselines by strategically leveraging unlabeled data, with success driven by feature quality (e.g., KeyLinker) rather than the volume of pseudo-labels.

**Weaknesses:**

W1. The technical contribution is limited. No new methodology, algorithms, systems are proposed. The novelty of the features is unclear, since they are derived from related work - KeyLinker Smolenkova & Yanovich (2025), an address clustering technique and enhanced Shared Send untangling metrics Larionov & Yanovich (2023).

W2. Beyond effectiveness, no other performance metrics are considered such as efficiency and scalability, interpretability, etc.

W3. The experiment is conducted on one dataset. How well the results generalize across multiple datasets is unclear. No specific case studies are demonstrated.

**Questions:**

The main bottleneck is the limited technical contributions. See all the concerns listed in Weaknesses.

---

### Meta-Review · Area_Chair_gqpH · 2026-01-05

**Summary:**

This paper studies semi-supervised detection of illicit Bitcoin flows through feature engineering. The authors release a large historical dataset comprising 163 million Bitcoin transactions with newly designed high-fidelity features, and empirically demonstrate that semi-supervised learning, when guided by these quality-focused features, can effectively leverage unlabeled data to achieve an F1 score of 0.84. However, all reviewers note that the paper’s technical novelty is limited and that the evaluation is insufficient. Additionally, the authors did not provide a rebuttal. I agree with the reviewers and suggest rejecting this paper.

**Reviewer Concerns:**

The authors did not provide a rebuttal. As a result, none of the reviewers’ concerns were addressed.

**Reviewer Scores:**

Since no response was provided to clarify or mitigate the identified weaknesses, the reviewer’s assessment and score would likely remain unchanged.

---

### Decision · Program_Chairs · 2026-01-26

Reject